# Total Style Transfer with a Single Feed-Forward Network

**DOI:** 10.3390/s22124612

**Published:** 2022-06-18

**Authors:** Minseong Kim, Hyun-Chul Choi

**Affiliations:** 1Alchera Inc., 225-15 Pangyoyeok-Ro, Bundang-gu Seongnam, Seongnam-si 13494, Korea; ms.kim@alcherainc.com; 2ICVS Laboratory, Department of Electronic Engineering, Yeungnam University, 280 Daehak-Ro, Gyeongsan-si 38541, Korea

**Keywords:** image style transfer, multi-scaled style transfer, intra-scale transformer, inter-scale transformer, computer vision, deep learning

## Abstract

The development of recent image style transfer methods allows the quick transformation of an input content image into an arbitrary style. However, these methods have a limitation that the scale-across style pattern of a style image cannot be fully transferred into a content image. In this paper, we propose a new style transfer method, named total style transfer, that resolves this limitation by utilizing intra/inter-scale statistics of multi-scaled feature maps without losing the merits of the existing methods. First, we use a more general feature transform layer that employs intra/inter-scale statistics of multi-scaled feature maps and transforms the multi-scaled style of a content image into that of a style image. Secondly, we generate a multi-scaled stylized image by using only a single decoder network with skip-connections, in which multi-scaled features are merged. Finally, we optimize the style loss for the decoder network in the intra/inter-scale statistics of image style. Our improved total style transfer can generate a stylized image with a scale-across style pattern from a pair of content and style images in one forwarding pass. Our method achieved less memory consumption and faster feed-forwarding speed compared with the recent cascade scheme and the lowest style loss among the recent style transfer methods.

## 1. Introduction

Since the development of neural style [1] using the VGG feature network [2], various image style transfer methods, such as photo-realistic style transfer [3], feed-forward networks [4,5,6,7,8,9], spatial color control [10] and fast and arbitrary style transfer methods [11,12,13] have been proposed. Subsequently, recent fast and arbitrary style transfer methods [11,12,13] demonstrated fast performance on transferring the style of an arbitrary style image into a content image by using feature transformers that transform the feature statistics of the content image into that of the style image.

However, these methods have a limitation in that multi-scaled style patterns of the style image cannot be integrally transferred to the content image. The first arbitrary style transfer method [11] cannot transfer the entire multi-scaled style patterns of style image to a content image because its AdaIN layer transforms only a single-scaled feature map of a content image. Multi-scaled feature transform methods [12,13] can transform intra-scale feature statistics of multi-scaled feature maps, which resulted in an incomplete transferring of multi-scaled style patterns.

These methods also used a scale-by-scale sequential transferring scheme, where the transferred style pattern of a scale is affected by the feature transform of the subsequent scales of the feature map because the multi-scaled feature map has inter-scale correlations. In this paper, we propose a new style transfer method, named total style transfer, which overcomes the current limitation of sole usage of intra-scale feature statistics of multi-scaled feature maps [11,12,13] by additionally using correlations between differently scaled maps.

As shown in Figure 1b, both intra and inter-scale correlations exist in the multi-scaled feature maps extracted from the image of Figure 1a. Therefore, we use a second order statistics, i.e., mean and intra/inter-scale covariance(s), to transform the intra/inter-scale feature statistics of multi-scaled feature maps of a content image into that of a style image. To generate a stylized image from transformed multi-scaled feature maps, instead of using sequential feature transform layers [11,12,13], we propose to use a single multi-scaled feature transform layer and a single network of U-Net [14] architecture using skip-connections as shown in Figure 2.

In our network, we transform the feature map once in a multi-scaled feature transformer and concatenate skip-connected feature maps with decoded feature maps to add the style of the current scale without losing the transferred style from the previous scale. In addition, to train the decoder network to generate multi-scaled stylized images, we improve the existing style loss [1,11] to consider the intra/inter-scale feature statistics consistently to our multi-scaled feature transform layer.

The contributions of our work are summarized as follows:We propose a new multi-scale feature transform layer where the intra/inter-scale feature statistics of a content image are transformed into that of a style image. As a result, the scale-across style patterns are transferred from the style image to the content image.We reduce multiple decoders and multiple feature transform layers into a single decoder and a single transform layer. As a result, our network quickly generates an output image through a single feed-forwarding pass.We propose a new style loss to learn a decoder network that utilizes both advantages of the existing style losses and intra/inter-scale feature statistics. This allows the decoder network to learn a consistent function to generate stylized images because our transform layer and style loss are consistent with each other.

In the rest of this paper, we review the existing style transfer methods in detail in Section 2, the proposed method is described in Section 3, the effectiveness of our method is demonstrated in Section 4, and the conclusion of this paper is presented in Section 5.

## 2. Related Works

Gatys et al. [1] proposed a neural style transfer using multi-scaled feature maps extracted from the VGG feature network [2]. They used a feature map from a certain deep layer to represent the content feature of an image and Gram matrices, i.e., correlations between channels of each single-scaled feature map to represent the style feature of an image.

To generate a stylized image, their method optimized a random noise or a content image pixel-wisely to minimize both content loss and style loss. The former is the difference between the content features of input content images and the latter is the difference between the style features of input and target style images. This method can transform an image into any target style; however, it takes a long time to be processed in real-time.

Soon after, several feed-forward networks [4,5,8,9] replaced the time-consuming online optimization process of neural style transfer [1] with an offline learning process. As a result, these methods can transfer the style of an image through a fast network feed-forwarding pass. However, there is a limitation in these methods in that only one style can be transferred per network.

The fast and arbitrary style transfer methods [11,12,13] have achieved fast style transfer as well as arbitrary style transfer overcoming the previous limitation of one style per network. Huang and Belongie [11] proposed an adaptive instance normalization (AdaIN) layer that adjusts the channel-wise mean and standard deviation of an encoded single-scaled feature map of a content image into that of a target style image. Although AdaIN can adjust the statistics of the encoded feature map, it has a limitation in that it does not consider the correlations between the feature map channels.

Li et al. [12] proposed whitening and coloring transform (WCT), which transforms the mean and covariance instead of the mean and standard deviation of AdaIN where covariance considers the correlations between feature map channels. To transfer the multi-scaled style of a content image, Li et al. [12] used a cascade of several transfer networks corresponding to each single-scaled feature map. Sheng et al. [13] used several sequential transform layers in a decoder network for a simple feed-forwarding process instead of using the network cascade. Since these multi-scaled style transfer methods [12,13] do not consider the existing correlations between different scales of feature maps (Figure 1b), their transferred style of large scale patterns can be distorted by the subsequent style transfer networks or transform layers of smaller scale patterns.

To consider the correlation between different scales of style, style loss [15] and feature transform methods [16] using cross-correlation between two feature maps of adjacent scales have been proposed. However, they have a limitation of no fully utilizing all intra/inter-scale feature statistics and only use the cross-correlation between adjacent scales.

## 3. Method

### 3.1. Intra/Inter-Scale Feature Transformer

In this section, we propose a new method that transforms a set of multi-scale feature maps to have the same intra/inter-scale correlations of a target set of multi-scaled feature maps. Here, intra-scale correlation is the correlation between two different channels of the feature map in each scale. The pixels on the diagonal rectangular regions of Figure 1b correspond to intra-scale correlation. Inter-scale correlation is the correlation between two channels of feature maps in different scales. The pixels on the off-diagonal rectangular regions of Figure 1b are corresponding to inter-scale correlation.

Intra-scale feature transform is a simple extension of the Single-Scale feature Transform (SST) method (CORAL [17] or WCT [12]) into a set of single-scale feature transforms with a transformer per scale. Suppose we have two feature maps $F_{x,i}\in R^{C_{i}\times \left(H_{x,i}\cdot W_{x,i}\right)}$ of a content (x=c) image and a style (x=s) image with number of channels ($C_{i}$), spatial height ($H_{x,i}$) and width ($W_{x,i}$) on the *i*-th scale. SST consists of two sequential processes, style normalization and stylization, to transform the style of the content image using the extracted feature maps $F_{x,i},\;x\in \left\{c,s\right\}$. In the style normalization step, we first calculate a zero-centered feature map ${\bar{F}}_{c,i}\in R^{C_{i}\times \left(H_{c,i}\cdot W_{c,i}\right)}$ from the content feature map $F_{c,i}$ along the spatial axis and then normalize it using its covariance, $cov\left({\bar{F}}_{c,i}\right)\in R^{C_{i}\times C_{i}}$, as in Equation (Equation 1).
(1)$${\hat{F}}_{c,i}=cov{\left({\bar{F}}_{c,i}\right)}^{-\frac{1}{2}}\cdot {\bar{F}}_{c,i}.$$

In the stylization step, the normalized content feature map ${\hat{F}}_{c,i}$ is stylized into $F_{cs,i}$ by the covariance, $cov({\bar{F}}_{s,i})$, and the mean vector, $\mu_{s,i}\in R^{C_{i}\times 1}$, of the style feature map $F_{s,i}$ as in Equation (Equation 2).
(2)$$F_{cs,i}=cov{\left({\bar{F}}_{s,i}\right)}^{\frac{1}{2}}\cdot {\hat{F}}_{c,i}+\mu_{s,i}\cdot 1^{1\times \left(H_{c,i}\cdot W_{c,i}\right)}.$$

To perform a multi-scaled feature transform (MST), we apply SST to the three content feature maps $F_{c,i}$ corresponding to the i=1{relu_1_2},2{relu_2_2},3{relu_3_3} layers of the encoder network as shown in Figure 3 Ours-intra.

For inter-scale feature transform to transfer scale-across style patterns to a content image, we additionally consider the correlations between the feature map channels of different encoder layers, where the feature map of each layer holds the responses of an input image to receptive fields of a certain scale.

First, we need to match the sizes of the feature maps of different scales to calculate the inter-scale correlation because they have different spatial sizes due to the pooling layers of the encoder network [2]. Not to lose any pattern and style information of all feature maps, we upsample all feature maps $F_{x,i}$ into the size of the largest feature map $F_{x,1}$ as shown in Figure 3a. Then, the feature maps of the same spatial size are concatenated into a multi-scaled feature map, $F_{x}^{\prime }\in R^{\left(\sum_{i}C_{i}\right)\times \left(H_{x,1}\cdot W_{x,1}\right)}$, along the channel axis as in Equation (Equation 3).
(3)$$F_{x}^{\prime }={\left[{F_{x,1}}^{T}\;\;{U(F_{x,2})}^{T}\;\;{U(F_{x,3})}^{T}\right]}^{T},\;x\;\in \;\left\{c,s\right\},$$
where *U* represents a function of upsampling into the size of $F_{x,1}$. This concatenated multi-scaled feature map goes through SST (WCT in Figure 3 Ours-inter), and we finally obtain a transformed feature $F_{cs}^{\prime }$ that has the same second-order statistics, i.e., the mean and covariance of the target style feature map up to inter-scale.

To generate a stylized image using the transformed feature map $F_{cs}^{\prime }$, we use the decoder network shown in Figure 2, which requires the stylized feature maps $F_{cs,i}^{\prime }$ in their original sizes. Therefore, the multi-scaled feature map $F_{cs}^{\prime }$ is sliced along the channel axis as shown in Figure 3b, and the spliced feature maps are downsampled to the original spatial sizes as in Equation (Equation 4):(4)$$F_{cs,i}^{\prime }=D_{i}\left(F_{cs}^{\prime }\left[\sum_{k}^{i-1}C_{k}+1\;:\;\sum_{k}^{i-1}C_{k}+C_{i}\right]\right)\;\;for\;\;i\geq 1,$$
where $D_{i}\left(f\right)$ is a function that downsamples the input *f* to the size of $F_{x,i}$.

The transformed feature maps $F_{cs,i}$ (Equation (Equation 2), intra-scale) or $F_{cs,i}^{\prime }$ (Equation (Equation 4), inter-scale) are inserted into the corresponding decoder layers through skip-connections to generate an output stylized image. The details of this process are described in Section 3.2.

### 3.2. Single Decoder Network with Skip-Connections

We need to utilize both a decoded feature map from the previous decoder layer and an intra or inter-scale transformed feature map from the transformer of Section 3.1 in each layer of the decoder network to generate the output stylized image considering intra or inter-scale correlations for all scales. For this purpose, we adopt skip-connections, which have been applied to several applications in the computer vision field [14,18,19] to merge two different feature maps in a decoding process. Skip-connected two-scale features are optimally merged by a trainable convolution layer, and this improves the quality of a decoded image by considering multi-scaled filter responses.

Unlike the existing applications [14,18,20] that skip-connected the originally encoded feature maps of an input image in the decoding process, we use the transformed feature maps of Section 3.1 in a decoder network. Our method is also different from the previous cascade scheme of [12] because we use a single encoder/decoder network, parallel transformers for each scale feature and merge multi-scaled styles optimally, while the cascade scheme requires several encoder/decoder networks (one network per scale) and sequentially transfers styles from the large to the small scale at the risk of degradation in the previously transferred scale of style.

Avatar-Net [13] also used a single decoder network and skip-connections; however, it sequentially applied feature transformers from large to small scale without considering possible degradation of the previously transferred scale and did not consider inter-scale correlations.

### 3.3. Intra/Inter-Scale Style Loss

To generate the stylized image with multi-scaled style patterns, in addition to a new multi-scaled feature transformer, a new style loss is necessary to train a style transfer network because the existing losses, such as Gram loss [1], Mean+Std loss [11] and Reconstruction loss [12,13] do not consider the multi-scaled style or correlations between styles of different scales. Therefore, we propose new style losses, Mean+Intra-cov and Mean+Inter-cov, that consider intra or inter-scale style correlations. Given the covariance matrix $cov({\bar{F}}_{x,i})$ and mean vector $\mu_{x,i}$ of the multi-scaled feature maps extracted from an output image (x=o) and a target style image (x=s), intra-scale style loss (Mean+Intra-cov, ${L_{style}}_{intra}$) is calculated as in Equation (Equation 5).
(5)$${L_{style}}_{intra}=\sum_{i}\left|\right|\mu_{s,i}-\mu_{o,i}\left|\right|+||cov\left({\bar{F}}_{s,i}\right)-cov\left({\bar{F}}_{o,i}\right)\left|\right|.$$

Given the multi-scaled feature map $F_{x}^{\prime }$ as described in Section 3.1, then its covariance matrix $cov({\bar{F^{\prime }}}_{x})$ and mean vector $\mu_{x}^{\prime }$, inter-scale style loss (Mean+Inter-cov, ${L_{style}}_{inter}$) are calculated as in Equation (Equation 6).
(6)$${L_{style}}_{inter}=\left|\right|\mu_{s}^{\prime }-\mu_{o}^{\prime }\left|\right|+||cov\left({\bar{F^{\prime }}}_{s}\right)-cov\left({\bar{F^{\prime }}}_{o}\right)\left|\right|.$$

These two losses are consistent with the feature transformers in Section 3.1 and using consistent pair of style loss and feature transformer, i.e., using Mean+Intra-cov loss with intra-scale feature transformer or using Mean+Inter-cov loss with inter-scale feature transformer, resulted in a better quality of style transfer in our experiments.

## 4. Experiments

### 4.1. Experimental Setup

To exclude any influence of network structure on image style transfer, we commonly used VGG16 feature network [2] as an encoder network and constructed a decoder network with the mirrored architecture of the encoder. Note that the number of convolution filter channels of the skip-connected layers in the decoder network was doubled. We used the feature maps of {relu_1_2, relu_2_2, relu_3_3, relu_4_3} to compute the style loss and used the feature map of {relu_3_3} to compute the content loss.

We used the training sets of MS-COCO train2014 [21] and Painter By Numbers [22] as a content image set and a large style image set, respectively. Each image set consists of about 80,000 images. To compare the effectiveness of the proposed method for varying numbers of style images, we also made a small style image set, which consisted of 77 style images collected from various sources. We used MS-COCO test2014 [21] and test images of Painter By Numbers [22] as test image sets.

As a common pre-processing, we resized the length of the short side of the training and test images to 256 pixels preserving their aspect ratios. Additionally, images were randomly cropped into 240 × 240 pixels to avoid boundary artifacts only in the training phase.

We trained the decoder network at a learning rate of $10^{-4}$ during four epochs with four randomly selected (content image, style image) batches. All experiments were performed with Pytorch framework on NIVIDA GTX 1080 TI GPU device.

### 4.2. Comparison in Output Style Quality

For comparing the stylization qualities of the existing methods and ours, we generated the result images of Neural Style [1] by performing 300 iterations using L-BFGS optimizer [23] with a learning rate of $10^{0}$, the result images of AdaIN [11] in the same environments as the proposed method except for feature transformer and loss. We used the style strength (α=0.6) proposed by [12] to generate the result image of Universal [12]. Furthermore, the patch size of 3×3 was used for feature swap in Avatar-Net [13].

The network architectures of the previous multi-scaled style transferring methods [12,13] were configured to use transformer on three scales as with our method. Figure 4 shows the result images of our method and the existing methods [1,11,12,13]. As shown in Figure 4, the image qualities of the previous fast and arbitrary style transfer methods using feed-forward networks (Figure 4b–d) were degraded compared with the results of pixel-wise optimization (Figure 4a).

In particular, the multi-scaled patterns of the style image were not transferred well. In the result images of our method (Figure 4e,f), particularly the first and third rows, the multi-scaled patterns of the style image transferred better than in the results of the existing methods (Figure 4b–d).

In the fifth row of Figure 4, the result image of successive transformers (Figure 4c,d) [12,13] could not preserve the transferred style from the previous scale and rather degraded the output style quality. Therefore, only a certain scale pattern (line patterns of a certain thickness) strongly presented with blurry spots in the result images. On the other hand, style patterns of various sizes were transferred to the result images of our methods (Figure 4e,f) with our multi-scaled feature transformer and intra/inter-scale style loss.

To quantitatively compare the performance of our method with the existing methods [1,11,12,13], we calculated the content and style losses of the result images generated from each method. Each value in Table 1 is the average (standard deviation) loss of 500 random test images. Neural Style [1] achieved the best performance in the content loss and all kinds of style losses as predicted by the plausible visual style quality in Figure 4. Since Universal [12] and Avatar-Net [13] trained their style transfer network with only reconstruction loss, they showed relatively higher content and style losses although their result images appeared to present the target style to some degree as shown in Figure 4c,d.

Our methods and AdaIN [11] achieved the first (red-colored values) or the second (blue-colored values) lowest values in style losses among the fast and arbitrary style transfer methods (Table 1 (b–f)) since they used style loss for network training. AdaIN achieved a lower Mean+Std style loss compared with Ours-intra because AdaIN used Mean+std style loss, while Ours-intra used Mean+Intra-Cov loss.

For the content loss, Ours-inter, which achieved the best style loss, showed relatively higher content loss compared with Ours-intra and AdaIN. This is because strongly transferring the target style to the result image led to some loss in content preservation. For instance, the images in the first and the last rows of Figure 4 show strong cell patterns of the target style on face or brick patterns on buildings, and this resulted in the unclear shape of face or building of the result images. This indicates that content preservation and style expression are a trade-off, and therefore Ours-inter with the lowest style loss had relatively high content loss. We can select Ours-intra for high content maintenance or Ours-inter for high style expression.

Figure 5 shows some result images of ours and the the existing style transfer methods [1,11,12,13] based on VGG 19 feature network [2]. The layer configurations of Neural Style [1], AdaIN [11], Universal [12] and Avatar-Net [13] are the same as those mentioned in each paper. Furthermore, the feed-forward networks of our methods consist of an encoder with up to {relu_4_1} of VGG 19 feature network (same to the single network methods [11,13]) and a decoder that has the mirrored structure of the encoder. As shown in Figure 5, we can see that our methods with VGG19 feature network also show better output style quality of multi-scaled pattern compared with the other previous methods as with the VGG16 feature network.

### 4.3. User Study

We performed a user study using the result images of the existing style transfer methods [1,11,12,13] and our proposed methods. First, we generated 66 stylized images for each method given all possible (content image, style image) pairs of six test content images and 11 test style images. Secondly, we randomly selected six images per method from the 66 stylized images without duplication of the content image As in [12], the output stylized images of all methods were placed side-by-side in random order.

Then, we asked each user to rank them based on how well the target style was applied to the output image. Finally, we had 246 votes from 41 people who did not have any prior knowledge of the image style transferring technique. From the voting results, we counted the number of top-1 votes for each method. Table 2 shows the results of our user study.

As a result, Ours-intra received the best top-1 votes (63 votes), and Neural Style received the second best. Based on these voting results (Table 2) and the loss values (Table 1), it is difficult to conclude that the user preference simply correlated with a loss measurement. For example, several methods with low style losses (Neural Style, AdaIN and Ours-intra) were preferred; however, other methods with low style losses (Avatar-Net and Ours-inter) were not, and even Universal with the highest style losses had many top-1 votes.

Another interesting result is that the voters tended to put more weight on content maintenance than on style expression. For example, Neural Style and Ours-intra, which achieved the lowest content loss received the highest top-1 votes, while Avatar-Net and Ours-inter of relatively high content loss recorded the lowest top-1 votes.

### 4.4. Comparison in Speed and Memory Efficiency

Here, we compared our method and the existing methods [1,11,12,13] in terms of the processing speed and memory consumption.

In the aspect of processing speed, the average (standard deviation) of the elapsed times for 500 test images of 256 × 256 pixels are presented in the second to fifth rows of Table 3. The feed-forward methods (Table 3 (b∼f)) were much faster compared with the Neural Style methods (Table 3 (a)) due to the heavy optimization process. The single-network-based methods (Table 3 (b,d∼f)) had a faster decoder speed compared with the cascade-network-based method (Table 3 (c)). In particular, the methods utilizing WCT [12], which requires singular value decomposition (SVD) of high computational cost, showed a large drop in speed (Table 3 (c∼f)) compared with AdaIN [11] using a light-weight feature transformer (Table 3 (b)).

The decoding speeds of our methods (Table 3 (e,f)) were slightly slower compared with the previous fastest methods of the same network architecture (Table 3 (b,d)) because of the doubled number of filter channels at the skip-connected layer as mentioned in Section 4.1; however, our methods achieved the lowest (Table 3 (e)) or similar (Table 3 (f)) total processing times compared to the previous fast and arbitrary style transfer methods using WCT (Table 3 (c,d)), the correlation-aware transform layer with a high computational cost.

In the aspect of memory consumption, we counted the total numbers of parameters in the networks of the methods and presented them in the last row of Table 3. Our methods (Table 3 (e,f)) had slightly larger numbers of parameters when compared with the previous methods of the same network architecture (Table 3 (b,d)) because the number of filter channels was doubled at the skip-connected layer of our methods as mentioned in Section 4.1. However, our methods used about 3% fewer parameters compared with Universal, which cascades several networks for multi-scaled stylization [12].

### 4.5. Multi-Scale Style-Content Interpolation

We compared the style-content interpolation results of Avatar-Net [13] and our methods (Ours-intra/inter). The result images were generated by varying the style-content trade-off factor *a* [11] of all multi-scaled transformers (for our methods) or a decorator (for Avatar-Net) in VGG 19 network architecture. Figure 6 shows the result images. When a=0.00 for content reconstruction, the images of our methods (Figure 6a,b)) were very similar to the content image but with a slightly darker tone, while the images of Avatar-Net (Figure 6c) had the color tone of the style image, which was different from that of the content image.

As Avatar-Net used decorator-only interpolation (a single-scale interpolation) and no interpolation in the AdaIN layers close to the output image, low-level features, such as the color tone of an image could not be adjusted to the content image. However, our method used multi-scaled style interpolation in every transform layer, and this resulted in a much better content reconstruction result.

Since reconstruction loss at the image level is known for better tone matching [12,13], we tested Ours-inter trained with the additional multi-scale reconstruction loss of Avatar-Net for better image tone matching. Figure 6d shows the results. Using the reconstruction loss, our method almost restored the content image when a=0.00 without losing the stylization ability when a=1.00.

### 4.6. Comparison in Characteristics

Table 4 shows the characteristics of the existing methods and ours. Neural Style [1] considered multi-scaled stylization and correlations but did not consider inter-scale style correlation in its style loss and was not operated in real-time. AdaIN [11] showed real-time performance but did not deal with multi-scaled stylization at all. Universal [12] is a real-time multi-scaled style transfer but uses a sequential transformer through cascade networks, and the transferred style of a scale is affected by the style transfer of the following scale.

Avatar-Net [13] used multi-scaled stylization with a single network but still used the sequential transformer scheme in its decoder and did not consider the intra-scale style correlation of low-level features and inter-scale style correlation at all scaled features. Our method has all of the previously improved characteristics and additionally considers intra-scale style correlation (Ours-intra), inter-scale correlation (Ours-inter) and multi-scaled style interpolation.

### 4.7. Comparison of Our Intra-Scale and Inter-Scale Stylizations

Feed-forward networks have their capacity of style representation, and we tested the style capacities of our Intra/Inter-scale methods by observing their result images from two decoder networks trained using a small style set and a large style set in Section 4.1. Figure 7 shows the result images. For both small and large numbers of styles, Ours-inter (Figure 7b,d) transferred the coarse to fine wrinkle patterns of the style image compared with Ours-intra (Figure 7a,c). This observation shows that the co-occurring multi-scaled patterns of the style image can be better transferred by using the inter-scale feature statistics.

Figure 8 and Figure 9 show the result images of Intra-scale and Inter-scale, respectively. The first row and column of each figure are the target content image and target style image, respectively, and all target images were not used during the network training phase. As we can guess from the loss values of Table 1, the images of Intra-scale (Figure 8) show better content-preserving results (lower content loss in Table 1), and the images of Inter-scale (Figure 9) show stronger and multi-scale style-presenting results (lower style loss in Table 1).

### 4.8. Performance of Our Intra/Inter-Scale Style Losses

To prove the effectiveness of our intra/inter-scale style losses independent of network architecture, we tested them with the direct pixel-wise optimization method [1]. Both Figure 10a,c show the results of style losses (Gram and Mean+Intra-cov) considering the channel correlation inside each scale and Figure 10b shows those of Mean+Std loss that do not consider channel correlation.

We can see that the curve pattern of the style image is transferred better to the result image with channel-correlation-aware loss (Figure 10a,c) compared with with Mean+Std loss (Figure 10b). When comparing the results with considering channel Mean in style loss (Figure 10b,c) and the result without channel Mean (Figure 10a), considering channel Mean has the effect of better transferring the average color tone of the style image to the result image.

Therefore, our Mean+Intra-cov and Mean+Inter-cov losses have the benefits of existing losses, i.e., the well-transferred style pattern by considering correlation (Gram) and the well-transferred average color tone by using Mean (Mean+Std).

### 4.9. Ablation Study of Skip-Connected Decoder Network

To verify the effectiveness of skip-connections (Figure 2) in generating result image, we compared the result images from three networks with the different number of skip-connections (0, 1 and 2) of Figure 11. When one skip-connection is utilized (Figure 11b), some multi-scaled patterns appear on the result image, and its color tone is slightly improved. The color tone of the result image becomes more similar to that of the style image as the second skip-connection is added (Figure 11c). We also observed that the more skip-connections used, the smaller the style loss of the result image, i.e., 0.34 (0.20) for one skip-connection to 0.31 (0.19) for two skip-connections. These style loss values are the averages (standard deviations) of Mean+Intra-cov losses of 500 test images.

To verify the effectiveness of skip-connections in the network training process, we analyzed the gradient ($\frac{dL}{dw}$) of total loss (*L*), i.e., the sum of the content loss and style loss, with respect to the convolution filter weights (*w*). The gradient values corresponding to decoded feature and skip-connected feature represent how strongly each feature influences loss reduction.

Figure 12 shows the absolute values of total loss gradient regarding to decoded feature map from decoder (channel index 0∼127 for Figure 12a or 0∼63 for Figure 12b) and skip-connected feature map from transformer (channel index 128∼255 for Figure 12a or 64∼128 for Figure 12b) in our network (Figure 2). Each value of the graph was sampled every 500 iterations. For the first skip-connection (Figure 12a), the absolute gradient corresponding to the skip-connected feature map is larger than that of the decoded one at the beginning of training process. This indicates that the feature map of the pre-trained encoder matters in reducing the total loss.

As the training iteration goes, we can see that the two feature maps have very similar gradient values, which shows the equal impact on reducing the total loss. For the second skip-connection (Figure 12b), there is almost no difference between the absolute gradient values corresponding to the two feature maps at the beginning of training. However, the absolute gradients corresponding to the decoded feature map become larger as the training process goes on. Since the skip-connected feature map of the previous scale has been accumulated into the decoded feature map, the decoded feature map becomes more important in reducing the total loss compared with a single-scale skip-connected feature map from the transformer. However, we also use the second skip-connection because this improves the output color tone as shown in Figure 11c.

## 5. Conclusions

In this paper, we proposed a new method called total style transfer for transferring multi-scaled patterns of a target style image to a content image. Our method improved the feature transformer and style loss of arbitrary style transfer to consider not only intra-scale but also inter-scale correlations between multi-scaled feature maps. We also proposed the use of a single decoder network using skip-connections with feature concatenation to generate stylized images using a set of transformed multi-scaled feature maps.

As a result, our method achieved transferring multi-scaled style patterns with the lowest style loss among the recent fast and arbitrary style transfer methods without losing memory and speed efficiency by using a single forwarding pass. Our method also made multi-scaled style-content interpolation possible and showed better results in content image reconstruction with the multi-scale reconstruction loss [13].

Additionally, our method can easily cooperate with end-to-end learning [24] or uncorrelated encoder learning [25] for faster speed. We can adopt the end-to-end learning scheme [24] to improve the style quality of the proposed methods. Figure 13 shows the result images of the proposed method with decoder-only learning and encoder/decoder learning (end-to-end learning scheme). The benefit of an end-to-end learning scheme is color tone improvement. For instance, the result images have an overall color tone that is more similar to the target style images as shown in Figure 13.

## Figures and Tables

**Figure 1 sensors-22-04612-f001:**
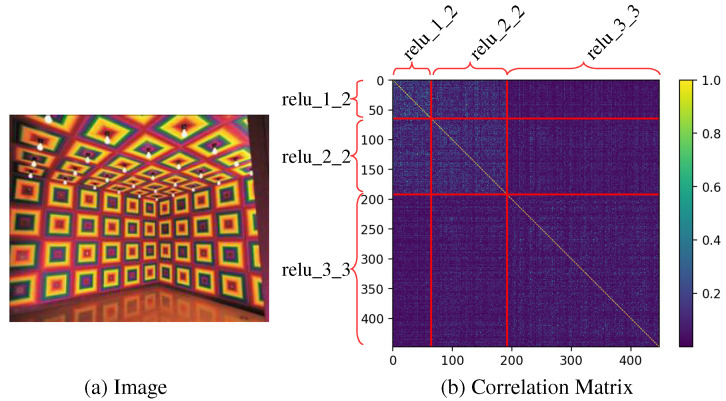
Correlation between channels (**b**) in the multi-scaled feature maps of an input image (**a**) extracted from the pre-trained VGG16 feature network [2]. The area corresponding to each scale of feature maps (relu_x_x) is divided by red lines. The diagonal rectangles of the correlation matrix (**b**) represent the intra-scale correlation between channels of the same scaled feature map (channels of relu_x_x), and the other rectangles represent the inter-scale correlation between the channels of two different scaled feature maps (channels of relu_x_x and relu_y_y).

**Figure 2 sensors-22-04612-f002:**
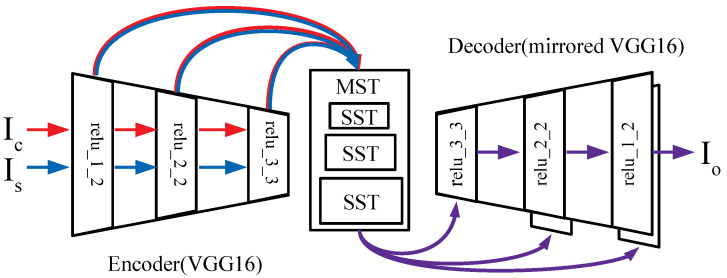
Network structure diagram for applying multi-scaled stylization: Our Multi-Scaled style Transformer (MST) transfers multi-scaled styles in a feed-forwarding process by using internal Single-Scale feature Transforms (SSTs) and a skip-connected decoder.

**Figure 3 sensors-22-04612-f003:**
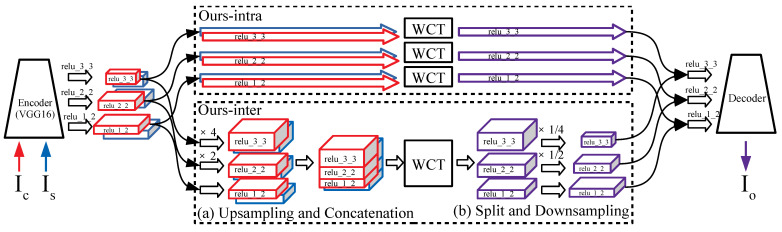
Each diagram shows the process of multi-scaled feature transform. Ours-intra presents multiple uses of SST to feature maps. Ours-inter represents merging multi-scaled features and dividing them into the original size for the inter-scale feature transform. (**a**) Merging multiple feature maps of several scales is performed as upsampling each scale feature by using nearest neighborhood interpolation to the largest size followed by concatenating them along the channel axis. (**b**) After feature transform, the transformed feature is split into multiple feature maps and downsampled into their original sizes. Each split feature is inserted into the decoder layer of the corresponding scale using skip-connection.

**Figure 4 sensors-22-04612-f004:**
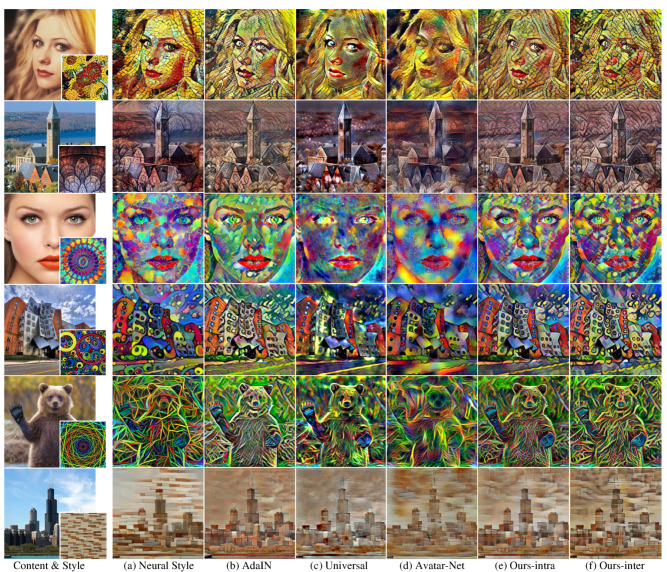
Output stylized images of the existing methods and our method with VGG16-based network structures. The target style images were not used for network learning.

**Figure 5 sensors-22-04612-f005:**
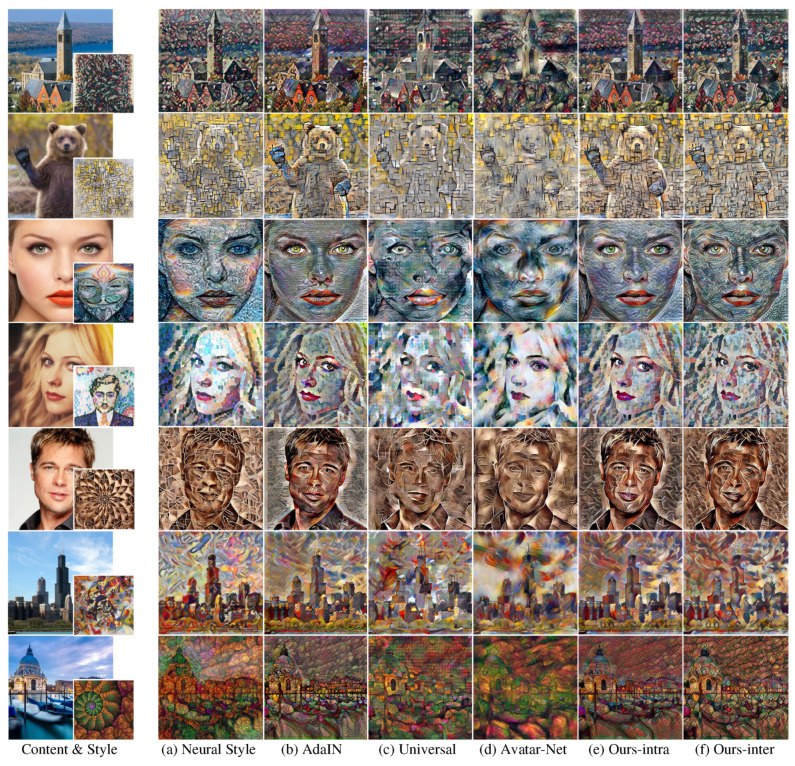
The result images of existing style transfer methods and Ours-intra/inter based on the VGG 19 feature network: The target content images and style images were not used during network learning.

**Figure 6 sensors-22-04612-f006:**
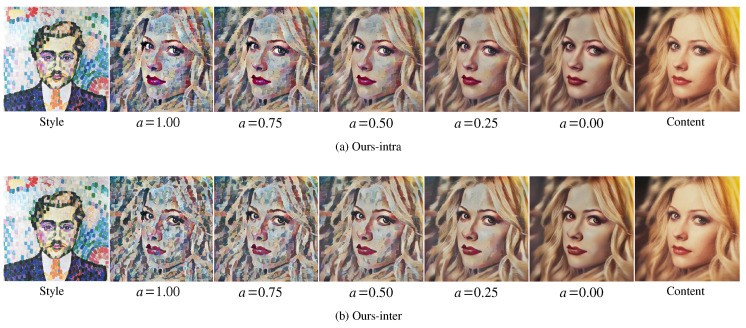
Style-content interpolation of our methods and Avatar-Net [13]. Avartar-Net cannot reconstruct content image when a=0.00, Ours-intra (**a**) and Ours-inter (**b**) achieved a smooth style interpolation from content image to style image with a slight tone mismatch when a=0.00. When Ours-inter (**d**) used additional multi-scale reconstruction loss of Avatar-Net, the image tone can be also matched to the content when a=0.00, while the original Avatar-Net (**c**) has some color bias.

**Figure 7 sensors-22-04612-f007:**
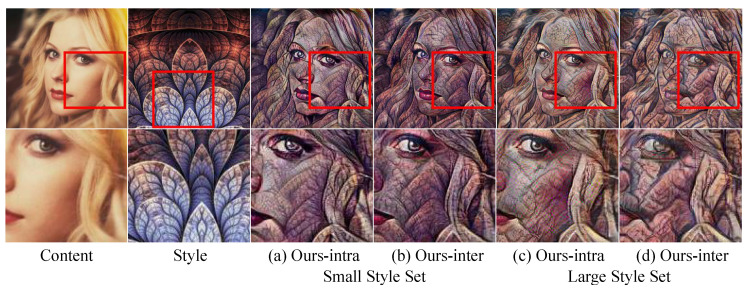
Comparison of intra-scale and inter-scale transform: (**a**,**b**) The results of networks trained with a small style set. (**c**,**d**) The results of networks trained with a large style set. (first row) Original images. (second row) The red box regions of the first row images.

**Figure 8 sensors-22-04612-f008:**
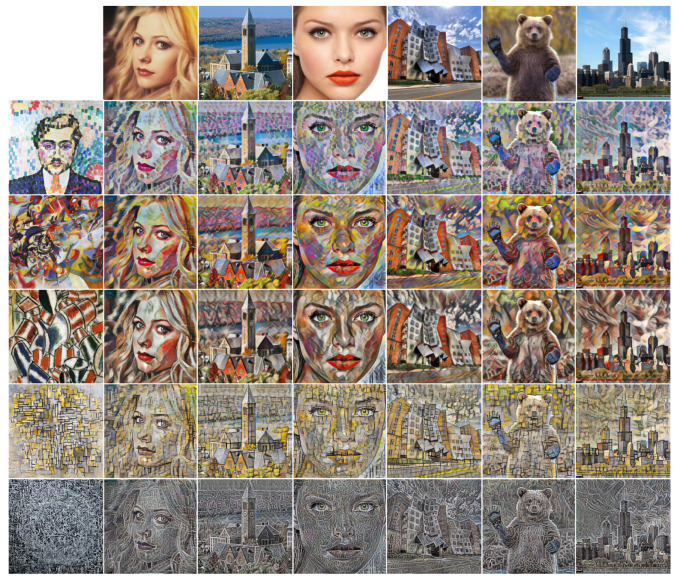
Additional result images of Ours-intra based on the VGG16 feature network: The first row and column are the target content image and target style image, respectively, and the target images were not used during network learning.

**Figure 9 sensors-22-04612-f009:**
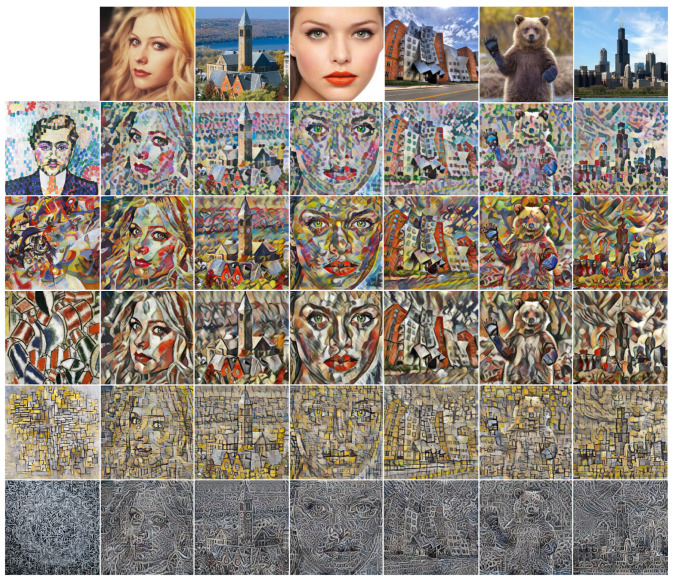
Additional result images of ours-inter based on the VGG16 feature network: The first row and column are the target content image and target style image, respectively, and the target images were not used during network learning.

**Figure 10 sensors-22-04612-f010:**
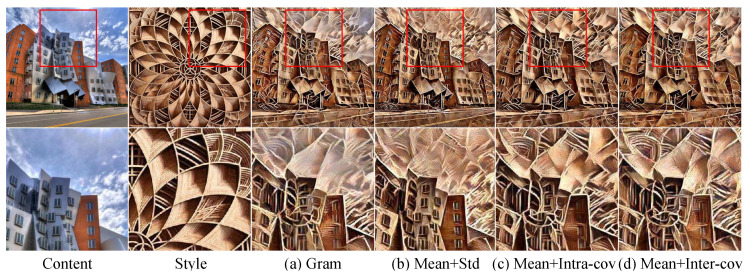
Comparison of style losses: each image is generated by using the specified style loss and the pixel-wise optimization method [1]. The images on the first row represent the original images and the image on the second row are the magnified images of the red box regions on the original images.

**Figure 11 sensors-22-04612-f011:**
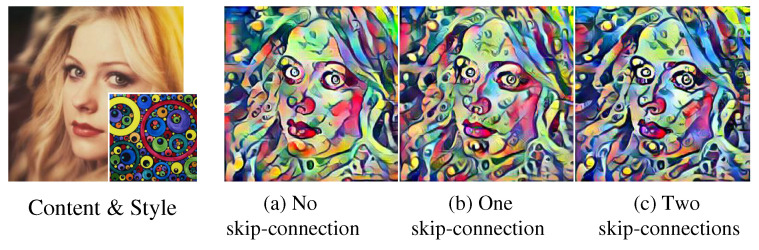
Output stylized images according to the number of skip-connections.

**Figure 12 sensors-22-04612-f012:**
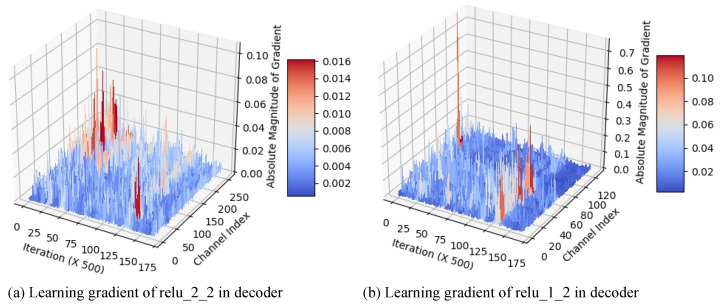
The amplitude of loss gradients with respect to the convolution weights in the skip-connected decoder layers during the learning process: The gradients are drawn every 500 iterations. The former half of the channels are for the decoded feature from the previous scale, and the latter half are for the skip-connected feature from the transformer.

**Figure 13 sensors-22-04612-f013:**
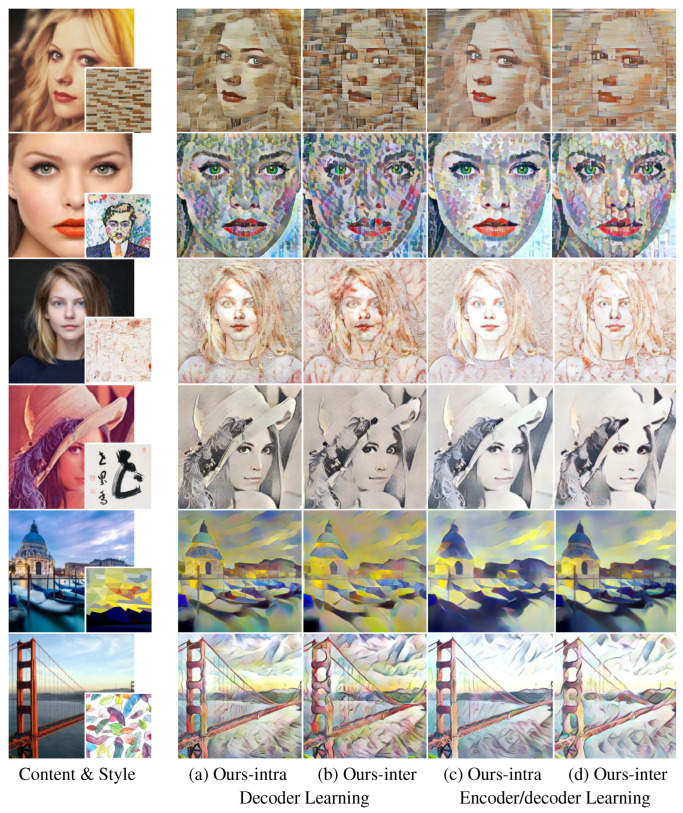
The result of the proposed method learning the networks with the end-to-end learning scheme and the proposed method learning the decoder network only: each target content image and style image were not used during network learning.

**Table 1 sensors-22-04612-t001:** Average (standard deviation) losses of various image style transferring methods. Red colored and blue colored values represent the best and the second best performances respectively.

	(a) Neural Style [1]	(b) AdaIN [11]	(c) Universal [12]	(d) Avatar-Net [13]	(e) Ours-Intra	(f) Ours-Inter
Content	**4.30 (2.30)**	**5.38(2.47)**	10.13(3.91)	8.92(2.65)	**4.68(2.04)**	7.59(2.61)
Style: Gram	**0.02 (0.01)**	0.28(0.90)	4.40(7.80)	0.52(1.90)	**0.26(0.79)**	**0.19(0.65)**
Style: Mean+Std	**0.04 (0.03)**	**0.19(0.16)**	2.56(2.51)	0.33(0.30)	0.23(0.16)	**0.15(0.13)**
Style: Mean+Intra-Cov	**0.13 (0.06)**	0.32(0.22)	2.25(1.61)	0.39(0.31)	**0.31(0.19)**	**0.25(0.18)**
Style: Mean+Inter-Cov	**0.15 (0.05)**	0.28(0.17)	3.90(2.69)	0.93(0.69)	**0.27(0.16)**	**0.23(0.15)**

**Table 2 sensors-22-04612-t002:** Top-1 votes of the style transfer methods. The red and blue colored values represent the best and the second best performances respectively.

Methods	Neural Style [1]	AdaIN [11]	Universal [12]	Avatar-Net [13]	Ours-Intra	Ours-Inter
# of Top-1 votes	**57**	43	54	2	**63**	25

**Table 3 sensors-22-04612-t003:** The average (standard deviation) elapsed time (ms) per each module and the number of total parameters in the image style transfer methods. The red and blue colored values represent the best and the second best performances respectively.

	(a) Neural Style [1]	(b) AdaIN [11]	(c) Universal [12]	(d) Avatar-Net [13]	(e) Ours-Intra	(f) Ours-Inter
Encoder time	-	1.14 (0.42)	2.25 (0.43)	**1.11 (0.17)**	**1.12 (0.22)**	**1.12 (0.21)**
Transformer time	-	**0.33 (0.15)**	90.70 (13.78)	208.77 (26.36)	**84.48 (15.17)**	101.74 (10.62)
Decoder time	-	**0.73 (0.39)**	1.52 (0.38)	**0.77 (0.19)**	0.93 (0.30)	0.89 (0.16)
Total time	172 K (203.01)	**2.20 (0.96)**	94.47 (14.59)	210.65 (26.62)	**86.53 (15.69)**	103.75 (10.99)
# of Parameters	-	**3471 K**	3769 K	**3471 K**	**3655 K**	**3655 K**

**Table 4 sensors-22-04612-t004:** Characteristics of the image style transfer methods.

	Neural Style [1]	AdaIN [11]	Universal [12]	Avatar-Net [13]	Ours-Intra	Ours-Inter
Real-time stylization	×	O	O	O	O	O
Multi-scale stylization	O	×	O	O	O	O
Independent scale style transfer	O	×	×	×	O	O
Intra-scale style correlation	O	×	O	O	O	O
Inter-scale style correlation	×	×	×	×	×	O
Multi-scale style interpolation	-	×	O	×	O	O
Single network	-	O	×	O	O	O

## Data Availability

Not applicable.

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
