# Peer review of "Total Style Transfer with a Single Feed-Forward Network"

_sensors, 2022, doi:10.3390/s22124612_

Round 1
Reviewer 1 Report
Review of article 1770943 sent to Sensors journal.
The authors propose a new paradigm to transfer the feature statistics of a content image following the ones of a style image using a multi-scale feature transformer, optimizing the resources for the transfer. They also propose a corresponding new style loss function.
After reading the article, I can mention that the research covers all aspects used in the style transfer CNN´s approaches, so I have no suggestions for improvement of the article. I have found only some writing errors; for example, on pages 2 and 9, some references have only the question mark, and also this sentence seems to be wrong written: Huang and Belongie [11] have proposed
Author Response
Thank you for your effort in reviewing our manuscript and your comments.
As you mentioned, we found the writing errors and reference errors. We have fixed those errors in the final manuscript.
Reviewer 2 Report
This article proposes a new style transfer method by employing intra/inter-scale statistics of multi-scaled feature maps.
The article is well organized and the topic should be quite interesting to the image processing and deep learning readers.
The authors compared the proposed methods with other methods in terms of output style quality, user assessment, speed and memory efficiency, etc.
There are lots of typos such as:
- fig. --> Fig.
- sec.?? --> Sec.
- eq. --> Eq.
This article needs proofreading. There are lots of "??"
Author Response
Thank you for your effort in reviewing our manuscript.
As you mentioned, we found many typos, writing errors, and reference errors. We have fixed those errors in the revised version.
Reviewer 3 Report
This paper presents a network for style transfer between images. One key contribution is the relatively lower computational complexity than the NeuroStyle while achieving a good performance compared to other similar networks.
Overall, the paper is well-written and explained. The results are convincing. Some comments to address:
- Some references to figures and sections are missing (they appear as ??)
- The complexity is measured in encoding/transformer/decoding times and no. of parameters. However, the times can be relative. The authors should include FLOPs as well to better understand complexity.
- One characteristic that is listed in Table 4 is real-time stylization. Can the authors provide more objective results to prove this is the case or explain what is meant by real-time in the case of working with images?
Author Response
Thank you for your effort in reviewing our manuscript.
As you mentioned, we found some typos and reference errors in our original manuscript. We have fixed them in the revised manuscript.
For the time complexity, since we have measured the timing values on the same device, we believe that the values can be an absolute measure of the time complexity of the models.
In table 4, real-time means the style transferring process can be done in a certain short time duration like several milliseconds enough to get 30 fps or higher when applying to video style transfer.